# Recent Advances in Cervical Cancer Management: A Review on Novel Prognostic Factors in Primary and Recurrent Tumors

**DOI:** 10.3390/cancers15041137

**Published:** 2023-02-10

**Authors:** Angela Santoro, Frediano Inzani, Giuseppe Angelico, Damiano Arciuolo, Emma Bragantini, Antonio Travaglino, Michele Valente, Nicoletta D’Alessandris, Giulia Scaglione, Stefania Sfregola, Alessia Piermattei, Federica Cianfrini, Paola Roberti, Gian Franco Zannoni

**Affiliations:** 1Pathology Unit, Department of Woman and Child’s Health and Public Health Sciences, Fondazione Policlinico Universitario Agostino Gemelli IRCCS, 00168 Rome, Italy; 2Anatomic Pathology Unit, Department of Molecular Medicine, University of Pavia and Fondazione IRCCS San Matteo Hospital, 27100 Pavia, Italy; 3Pathology Unit, Cannizzaro Hospital, 95126 Catania, Italy; 4Department of Surgical Pathology, Ospedale S. Chiara, 38122 Trento, Italy; 5Pathology Institute, Catholic University of Sacred Heart, 00168 Rome, Italy

**Keywords:** cervical cancer, prognosis, grading, pathology, outcome

## Abstract

**Simple Summary:**

The aim of the present review is to analyze the novel and most relevant prognostic factors in primary and recurrent cervical cancer. Based on our findings, tumour budding and cell nest size grading system, depth of stromal invasion, lympho-vascular space invasion, perineural invasion, tumor free distance and tumor infiltrating lymphocytes appeared the most relevant factors which may be included in the pathology report to help future studies to further elucidate cervical cancer prognosis.

**Abstract:**

Background: Several pathological parameters, including tumor size, depth of stromal invasion, lympho-vascular space invasion and lymph node status, have been proposed as prognostic predictors in cervical cancer. However, given the high mortality and recurrence rate of cervical cancer, novel parameters that are able to provide additional prognostic information are needed in order to allow a better prognostic stratification of cervical cancer patients. Methods: A search was conducted on PubMed to identify relevant literature data regarding prognostic factors in cervical cancer. The key words “cervical cancer”, “prognostic factors”, “pathology”, and “outcome” were used. Results: The novel pathological grading system based on tumor budding and cell nest size appeared the most relevant prognostic factor in primary neoplasms. Moreover, other potentially useful prognostic factors were tumor size, depth of stromal invasion, lympho-vascular space invasion, perineural invasion, tumor-free distance and tumor-infiltrating lymphocytes. Prognostic factors related to advanced-stage cervical cancer, including lymph-nodes status, endometrial and cervical involvement as well as distant metastases, were also taken into consideration. Conclusions: According to our findings, tumor budding and cell nest size grading system, depth of stromal invasion, lympho-vascular space invasion, perineural invasion, tumor-free distance and tumor-infiltrating lymphocytes appeared the most relevant factors included in the pathology report.

## 1. Introduction

Cervical cancer is one of the most common cancers worldwide, ranking as fourth for both incidence and mortality among all gynecological malignancies [1]. Squamous cell carcinoma (SCC) is the most frequent histotype, followed by adenocarcinoma (AC), which accounts for approximately 10–25% of cervical tumors [1,2].

Despite the majority of cases, especially in developing countries, being diagnosed at an advanced stage, an increasing percentage of tumors are diagnosed at an early stage [1,2]. For early-stage disease, several pathological parameters, including tumor size, histotype, depth of stromal invasion, lympho-vascular space invasion (LVSI) and lymph node status have been proposed as prognostic predictors, capable of stratifying patients into different risk groups [3,4,5,6].

However, given the high mortality and recurrence rate of cervical cancer, the abovementioned prognostic factors are still of limited value and provide suboptimal prognostic stratification for recurrence [1,3]. Therefore, novel parameters, able to provide additional prognostic information, are needed in order to allow a better clinical stratification of cervical cancer patients.

Actually, the histopathology report for cervical carcinoma should include all relevant information required for diagnosis, staging, prognosis and patient management. According to the most recent recommendations from College of American Pathologists (CAP) and the International Collaboration on Cancer Reporting (ICCR) [4], the following essential items should be included in the pathology report:-macroscopic tumor site-tumor dimensions (measurements of horizontal extent and depth of invasion or tumor thickness)-maximum and minimum length of vaginal cuff and parametria in two dimensions-histological tumor type and tumor grade-coexistent pathology (squamous intraepithelial lesion, adenocarcinoma in situ, stratified mucin-producing intraepithelial lesion)-LVSI-minimum distance of uninvolved cervical stroma-extent of invasion (vaginal, uterine corpus, parametrial, adnexal, bladder, rectum involvement)-margin status (for invasive tumors and for precursor lesions)-lymph node status: sentinel lymph node status, total number of nodes retrieved, number of positive lymph nodes-pathologically confirmed distant metastases.

Moreover, the following desired/recommended items should also be included in the pathology report:-HPV dependent and independent status-Silva pattern of invasion-Ancillary studies (p16 immunohistochemistry; in-situ hybridization for HPV).

The aim of our review is to identify and define novel and potentially useful clinicopathological prognostic factors for cervical cancer in order to provide a more accurate prognostic stratification in primary and recurrent tumors.

## 2. Materials and Methods

In this review we searched for previously published papers regarding pathological prognostic factors in cervical cancer, with a particular focus on those factors not yet codified in the current histological reporting guidelines. A search was conducted on PubMed to identify relevant literature data. The key words “cervical cancer”, “prognostic factors”, “pathology” and “outcome” were used.

## 3. Prognostic Factors in Primary Neoplasm

The most relevant prognostic factors in primary tumors are summarized in Table 1 and Table 2.

### 3.1. HPV Status

HPV infection represents the main pathogenetic event leading to cervical cancer development. The key step in the pathogenesis of cervical carcinomas is the integration of the HPV genome into the host chromosome, followed by the inactivation of viral E1 and E2 regions and upregulation of oncogenes E6 and E7 [7]. In detail, E6 oncoprotein degrades p53, inhibiting apoptosis while E7 protein stimulates cell proliferation by suppressing RB1 [7]. Despite the vast majority of cervical epithelial tumors being related to HPV infection, it is now recognized that a proportion of these tumors, mainly represented by adenocarcinomas, are not associated with HPV infection and carry more aggressive clinical behavior than HPV-related carcinomas [2,4]. In this regard, the 2020 WHO Classification of female genital tumors introduced a novel classification for cervical epithelial tumors based on the presence or absence of HPV infection [2]. In detail, adenocarcinomas are now categorized into HPV-associated (HPVA) and HPV-independent tumors (HPVI). This latter group includes the following histotypes: gastric-type, clear cell, mesonephric and endometrioid carcinoma [2,4]. On the other hand, the 2020 WHO Classification categorizes squamous epithelial tumors into HPV-associated and HPV-independent categories [2]. HPV-independent squamous cell carcinoma is exceedingly rare and shows a higher rate of lymph node metastases, with a consequent reduced disease-free and overall survival compared with HPV-associated carcinoma [8]. However, currently there are not yet differences in treatment strategies between HPV-associated and HPV-independent tumors.

The same distinction (HPV-dependent/HPV-independent) is applied for premalignant precursor lesions, squamous and glandular in type:-HPV-dependent SILs (squamous intraepithelial lesions): high grade and low grade, respectively corresponding to CIN1 and CIN2-3 dysplasia-HPV-dependent AIS (adenocarcinoma in-situ) and its SMILE variant (stratified mucin producing intraepithelial lesion)-HPV-independent AIS: gastric-type AIS and ALEGH (atypical lobular endocervical glandular hyperplasia)

According to ICCR criteria, in the morphological assessment of a cervical cancer, coexisting precursor lesions should always be mentioned; a pathologist should also document the involvement and the distance from the resection margins (ectocervical/vaginal cuff; endocervical; radial/deep stromal) since the presence of a positive margin may influence clinical management and follow-up [4].

To date, the detection of HPV infection on cervical cancer and precancerous lesions in formalin fixed-paraffin embedded (FFPE) tissues is mainly based on p16 immunohistochemistry, which is widely used as a surrogate marker for high-risk human papillomavirus (hrHPV) infection [9]. P16 positivity is defined as strong, block-type staining pattern, involving all squamous epithelium layers [4,9].

Direct HPV testing with RNA in-situ hybridization is generally preferred in selected cases such as tumors with morphologic features suggestive of HPV infection but lacking block-type staining for p16 or, alternatively, tumors/premalignant lesions with block-type staining for p16 but lacking the morphological features indicative of viral infection [4].

### 3.2. Grading of Cervical Cancer

To date, there is no widespread consensus regarding the prognostic significance of tumor grade, and no validated grading systems are currently available for cervical cancer [4,10,11,12,13,14,15,16,17,18,19]. Although tumor grading is considered a recommended (not required) pathological feature, in the recent recommendations of the European Society of Gynaecological Oncology (ESGO), the European Society for Radiotherapy and Oncology (ESTRO) and the European Society of Pathology (ESP), it is not taken into account in clinical management of the cervical cancer patients to assess the need for adjuvant therapy following surgery [10,11,12,13,14,15,16,17,18,19]. Similarly, the recent ICCR data set for the reporting of cervical cancers and the “Sedlis Criteria” do not take into consideration grading for adjuvant treatment algorithms [4].

#### 3.2.1. Squamous Cell Carcinoma

Several grading systems are currently applied to grade cervical squamous cell carcinoma [12,13,14,15,16,17,18]; these include:-the Broder’s system, based on the degree of keratinization, cytological atypia and mitotic activity;-the grading of invasive tumor front or the pattern-type of invasion (pushing versus infiltrative);-the typology of neoplastic cells and the presence/absence of keratinization (large-cell keratinizing, large-cell non-keratinizing, and small-cell non-keratinizing categories);-the WHO proposal that considers the degree of keratinization, nuclear pleomorphism, size of nucleoli and mitotic index.

Other authors have described more complex multi-factor grading systems in cervical squamous cell carcinomas, including both conventional grading tumor-related parameters and other factors such as depth of invasion, LVSI and host/stromal inflammatory reaction [19].

Recently, Jesinghaus et al. demonstrated the prognostic value of a novel pathological grading system in cervical squamous cell carcinoma based on tumor-budding and cell nest size [20].

The prognostic significance of this novel grading scheme has already been documented in squamous cell carcinomas of the lung, oral cavity, oesophagus and gastrointestinal tract [21,22,23,24]; whereas only a few but promising articles have been recently published on the prognostic role of tumor-budding and cell nest size in both squamous cell carcinoma and adenocarcinoma of the cervix [25].

Tumor buds are strictly related to invasiveness, motility and epithelial/mesenchymal transition and are generally defined as isolated cells or clusters of 4 or 5 cells in intratumoral or peritumoral areas (Figure 1A) [20,21,22,23,24,25]. Cell nest size is another parameter that provides a qualitative measurement of cellular dissociation; its prognostic significance has already been demonstrated for oral, pulmonary and oesophageal tumors [20,21,22,23,24,25]. In the article by Jesinghaus et al., two cohorts of cervical squamous cell carcinomas patients were graded using this novel grading system [20]. Tumor-budding activity was considered as low (1–14 budding foci in 10 high power fields) and high (15 or more budding foci in 10 high power fields). Cell nest size was evaluated as follows: large (cell nests comprising more than 15 cells); intermediate (cell nests comprising 5–15 tumor cells); small (cell nests of 2–4 tumor cells) (Figure 1B); single-cell invasion (singular, discohesive tumor cells without nested architecture). In particular, the smallest identifiable cell nests were used in scoring. Finally, a score was attributed to both budding activity (1–3 points) and cell nest size (1–4 points); the sum of both scores results in a 3-tier grading system: G1 (score ranging from 2 to 3); G2 (score ranging from 4 to 5); G3 (score ranging from 6 to 7). By utilizing this novel grading, a significant prognostic impact has been demonstrated on overall survival (OS) and disease-free survival (DFS). Moreover, G2/G3 tumors were significantly associated with higher tumor stage, LVSI, perineural invasion and nodal metastases. Interestingly, data from Jesinghaus et al. have been further validated in a recent paper by Zare et al., which evidenced a significant association of the new grading system with OS, DFS, higher tumor stage and lymph node metastases [26].

#### 3.2.2. Adenocarcinoma

Several studies suggest a grading system for HPVA adenocarcinomas based on a combination of architectural and nuclear features and similar to the FIGO grading system applied for uterine endometrioid carcinomas [27,28]. The most commonly used cut-offs for solid architecture set at ≤10% (grade 1), 11% to 50% (grade 2), and >50% (grade 3) has been recommended, due to its good prognostic significance. Tumors can be upgraded in the presence of marked nuclear atypia in the majority of cells (>50%) [7]. Moreover, a clearly defined subset of endocervical adenocarcinomas should be considered intrinsically high-grade regardless of the morphology. Most of these represent HPVI adenocarcinomas (gastric type adenocarcinoma, clear cell carcinoma and mesonephric adenocarcinoma).

Among the cervical HPVA adenocarcinomas, the following variant should be not graded because it is considered automatically high-grade: micropapillary carcinomas, mucinous adenocarcinomas and neuroendocrine carcinomas (also in the mixed forms, irrespective of the percentage of the neuroendocrine component) [29,30,31]. A recent paper by Shi et al. demonstrated the reliability of the grading scheme based on tumoral budding and small nest size also in endocervical adenocarcinomas where it seems to outperform the conventional Federation of Gynecology and Obstetrics (FIGO) grading and Silva pattern classification [32]. Therefore, if further studies on larger cohorts will show similar results, the novel grading system could be included in the pathology reports as an additional tool to guide the therapeutic management of cervical cancer patients.

### 3.3. Silva Pattern of Invasion for HPV-Associated Adenocarcinomas

Recently, the Silva Pattern Classification has been shown to correlate with the risk of lymph node metastasis and patient survival [4,33]. The Silva classification can only be applied to HPVA cervical adenocarcinoma and sub-classifies tumors into three patterns (A, B, C) based on the presence and degree of destructive stromal invasion, LVSI and grade of cytologic atypia [33].

In detail, Pattern A tumors are composed of well-formed glands without evidence of destructive stromal invasion, single cells, solid growth, high grade cytology or LVSI. Pattern B tumors show limited destructive invasion with individual cells or clusters of tumor cells not exceeding 5 mm in maximum diameter. Pattern C tumors are characterized by diffuse destructive invasion associated with desmoplastic reaction.

The current literature evidence, mainly based on retrospective studies, suggests that Pattern A tumors do not develop lymph node metastases and carry a very limited risk of recurrence; therefore, they can be suitable for conservative treatment without lymph node dissection [2]. The risk of lymph node metastases is very low for Pattern B adenocarcinomas, which may benefit from SLN mapping, especially if LVSI is present. Finally, Pattern C tumors have a more significant risk of nodal metastases and tumor recurrences; therefore, standard surgical treatment, including lymph node dissection, is more appropriate for these latter patients [2].

### 3.4. Lympho-Vascular Space Invasion (LVSI)

LVSI assessment is a required item in the pathology report of cervical cancer since it is one of the criteria used to select patients suitable for surgical radicality and adjuvant treatment [11]. Several studies have investigated the prognostic role of LVSI and its association with nodal and distant metastases and patient survival [11,34,35,36,37,38,39]. However, results are extremely heterogeneous since some studies have shown the negative prognostic role of LVSI while other studies failed to demonstrate statistically significant results [11,37,38,39]. This discrepancy across studies may be explained by the qualitative method utilized to assess LVSI: present or absent. In this regard, literature data in endometrial cancer patients demonstrated that a semi-quantitative evaluation may better stratify patient prognosis [36,37,38,39]. In detail, according to the “three-tiered approach” for endometrial cancer, LVSI has been classified as follows: (i) Absent: No LVSI; (ii) Focal: single focus of LVSI around the tumor; (iii) Diffuse: more than 1 focus around the tumor [40,41,42,43]. With this approach, a diffuse pattern of LVSI has been demonstrated as an independent prognostic factor for nodal metastases, recurrence and decreased survival; on the other hand, endometrial cancer patients with focal LVSI showed a significantly better outcome [40,41,42,43]. Regarding LVSI in cervical cancer, a recent study by Ronsini et al. demonstrated, for the first time, that a semi-quantitative evaluation of LVSI in early-stage cervical cancer patients could provide a more accurate survival stratification [43]. In detail, different clinico-pathological features and survival outcomes were observed in patients with absent, focal and diffuse LVSI, respectively. Moreover, diffuse LVSI was associated with increased risk of nodal metastases, parametrial involvement and positive surgical margins [43]. Literature data also showed that only LVSI outside the tumors border, so called satellite LVSI rather than intratumor LVSI, has a significant prognostic value in cervical cancer [44].

If future studies on large series will support these findings, a semi-quantitative evaluation of LVSI could be recommended in the pathology reports of cervical cancer patients in order to optimize the diagnostic and therapeutic process.

### 3.5. Perineural Invasion (PNI)

PNI, according to the Liebig criteria, is defined as the presence of tumor cells along the nerve circumference or invading any of the three layers of the nerve sheath (epineurium, perineurium and endoneurium) (Figure 2) [45]. PNI is frequently detected in several malignancies, including head and neck squamous cell carcinoma, colorectal adenocarcinoma, prostate cancer, cholangiocarcinoma and pancreatic cancer [45,46,47,48,49,50]. In cervical cancer, the reported incidence of PNI ranges from 7.0 to 35.1%; moreover, PNI is frequently detected in combination with other risk factors, such as LVSI, deep cervical invasion, large tumor size, tumor extension to the uterus, positive surgical margins, parametrial invasion and pelvic lymph node metastases [50,51,52,53]. Therefore, patients with PNI are more likely to receive adjuvant radiotherapy or concurrent chemo-radiation after surgery. However, the real prognostic impact of PNI in cervical cancer is poorly understood and is still a matter of debate. In detail, some studies demonstrated the role of PNI as independent prognostic factor for OS; other studies showed a significant correlation of PNI with DFS and OS at univariate but not at multivariate analysis, whereas other authors failed to demonstrate any prognostic role of PNI [50,51,52,53]. Despite its limited prognostic role, it is well known that PNI is frequently related to other poor prognostic factors, such as LVSI, deep stromal invasion, large tumor size and parametrium invasion; therefore, its real impact on prognosis needs to be better elucidated [50,51,52,53]. According to literature data, PNI could be considered as an intermediate-risk factor for cervical cancer patients that may aid in the selection of the more appropriate therapeutic approach [50,51,52,53].

### 3.6. Depth of Stromal Invasion (DOI)

Depth of stromal invasion (DOI) represents an essential tool to be included in the pathology report, not only for staging purposes but also for its potential role as prognostic factor in cervical cancer [5]. According to Sedlis criteria, DOI is expressed as inner third, middle third and outer third of cervical wall thickness infiltration [54]. Several studies showed that DOI represents an independent prognostic factor for OS and DFS and is strictly related to local recurrences. Moreover, a significant difference of prognosis has been demonstrated between tumors with full-thickness invasion and tumors reaching the cervical–parametrial transition zone [54,55]. According to recent studies, DOI may represent a reliable method to categorize the pathological tumor response in cervical cancer after neoadjuvant therapy [5,55]. In detail, a recent meta-analysis evidenced a statistically significant difference in survival between residual tumor with stromal invasion > and <3 mm. Therefore, a cut-off of 3 mm of residual stromal invasion seems to outperform all other residual tumor scoring systems for prognostic stratification of post-neoadjuvant treatment cervical cancer [5]. Moreover, the objectivity of the measurement of the depth of stromal invasion makes this system heavily reproducible with limited inter-observer variability.

### 3.7. Maximum Horizontal Extent of Tumor

The horizontal extent of the tumor represents the longitudinal extent if the tumor is measured in the superior–inferior plane, or the circumferential extent if the tumor is measured perpendicular to the longitudinal axis of the cervix. It is best calculated histologically for smaller neoplasms or grossly for larger tumors [4].

Despite literature data suggesting its potential role as independent predictor of survival in cervical carcinoma, it is no longer utilized to stage microscopic (Stage IA) disease [4,56]. Therefore, the horizontal extent of the tumor is now considered as an optional element and its inclusion in the pathology report is encouraged to:-give a more complete picture of tumor extent (length and width);-appreciate tumor volume;-help future studies to further clarify its prognostic role.

### 3.8. Parametrial Involvement

Strictly related to DOI, parametrial involvement is another important histological feature for pathological staging. Parametrial involvement represents an independent predictor of recurrence and shorter DFS for both squamous cell carcinoma and adenocarcinoma [57,58,59]. Therefore, infiltration of parametrial structures must be included in the pathology report along with an accurate measurement of the lateral extent of each parametrium [4].

### 3.9. Tumor-Free Distance (TFD)

TFD has been proposed as another potentially useful prognostic parameter [60,61,62,63]. TFD represents the minimum distance of uninvolved stroma between the tumor and peri-cervical stromal ring [60,61,62,63]. It is well-known that the risk of pelvic lymph node metastases, poor prognosis and poor survival is inversely related to the thickness of the remaining fibromuscular cervical stroma around the tumor [61]. In a recent study by Cibula et al., TFD, assessed at pre-operative imaging (MRI or US), outperformed other prognostic markers, such as tumor size or depth of stromal invasion, which does not take into account the size of the cervix and tumor location in the cervix [63]. This study established the best cut-off value of TFD at 3.5 mm and provided further evidence for the independent prognostic role of TFD [63]. Several studies have also attempted to determine the best threshold for histologically assessed TFD; reported cut-offs range from 2.5 mm to 5 mm, but a consensus has not been established yet [60,61,62,63]. The prognostic impact of histologically assessed TFD has recently been investigated by Bizarri et al. In this study, patients with TFD <3.0 mm showed a worse DFS and OS compared to patients with TFD >3.0 mm. Therefore, authors concluded that a TFD ≤3.0 mm represents a poor prognostic factor, especially in “low-risk factors” patients, related to increased risk of recurrence and lymph node metastases [60].

Accordingly, if further studies will validate these findings, TFD could represent a novel prognostic marker for pre-operative assessment of risk factors and for guiding adjuvant treatment decisions.

### 3.10. Tumor-Infiltrating Lymphocytes (TILs)

TILs are strictly related to the dynamic interaction of the host immune system to tumor antigens and to the tumor microenvironment. Evaluation of TILs, quantification and immunophenotyping can provide useful information for tumor progression and treatment strategies [64]. So far very few clinical studies have been conducted on the prognostic and predictive role of TILs in cervical cancer [65,66,67,68,69]. In detail, stromal TILs are more useful and have superior predictive value than intraepithelial TILs in terms of prognosis in squamous cell carcinoma [65]. Moreover, several TIL populations have been described, such as CD4+ and CD8+ T cells, Th17, γδ T cells, natural killer cells, Treg cells, B cells and macrophages [63,69]. Concerning their prognostic value, a higher prevalence of CD4+ and CD8+ seems to correlate with a better outcome; therefore, an accurate definition of TILs may provide useful information for both patient prognosis and therapeutic management [69]. Moreover, recent studies demonstrated that TILs may represent one of the most effective and specific adaptive cell therapies (ACT) in several tumors including cervical cancer [67,68]. The main goal of TILs therapy is to restore anti-tumor immunity by an in vitro selection of immune cell populations that are tumor specific, such as CD4+ and CD8+ [67,68]. In detail, CD8+ TILs have been shown to directly kill tumor cells when presented with neoantigens and are also capable of activating apoptosis-inducing FAS–FASL pathways; CD4+ T cells play a regulatory role by differentiating into different phenotypes, such as Th1, Th2, Th17 and Treg [67,68]. Clinical application of TIL therapy in cervical cancer is still at its initial stages; however, promising results, especially in HPV-related tumors, have been demonstrated since the selection of viral antigen-specific TILs, such as HPV E6/E7, have been shown to achieve a highly selective tumoricidal effect [67,68]. These preliminary findings are worthy of further studies in order to introduce this novel therapeutic regimen in the clinical practice.

### 3.11. Margin Status

According to ICCR guidelines, the status (positive/negative and the tumor distance) of all surgical resection margins (ectocervical, endocervical, radial/deep stromal and vaginal cuff) should be recorded [4]. The prognostic impact of close margins on local and overall recurrence in cervical cancer patients undergoing radical hysterectomy has been assessed by some retrospective studies [70].

According to Viswanathan et al., the local recurrence rate was 20% in FIGO Stage IB carcinomas with “close” margins (<10 mm) vs. 11% in patients with negative margins (≥10 mm) [71]. Moreover, McCann et al. suggested that close surgical margins (≤5 mm) were associated with recurrence rates of 24% as compared with recurrence rates of 9% in patients with negative margins [72].

## 4. Prognostic Factors in Advanced-Stage

The most relevant prognostic factors in advanced-stage tumors are summarized in Table 3.

### 4.1. Local Involvement: Endometrial, Adnexal and Vaginal Extension

The revised 2018 FIGO staging does not take into account cervical cancer extension in the endometrium, fallopian tubes, as well as superficial spread to the ovaries [4,11,73,74,75]. However, it is not uncommon for cervical cancer to involve these sites.

The true rate of lower uterine segment involvement is not well-established as this finding is not routinely described in the pathological report. Both adenocarcinomas and squamous cell carcinoma may show diffuse endometrial and myometrial involvement and more rarely an exclusive mucosal colonization of the endometrial glands [75]. Despite the prognostic significance of these cases remaining uncertain, it is important to take into account the possibility of this phenomenon in order to avoid an erroneous diagnosis of primary endometrial adenocarcinoma with cervical extension [76,77,78]. Uterine corpus invasion by cervical cancer is found in approximately 5% of patients, and it has been associated with significantly lower 5-year and 10-year survival rates compared with cervical cancer without uterine corpus invasion [76]. Other studies have described association with an increased risk of pelvic lymph node metastases and ovarian metastases [77,78]. In their multicenter retrospective study on stage IA2–IIB cervical cancer, Fangjie He et al. demonstrated that myometrial invasion ≥50% within the uterine corpus was associated with worse prognosis, while myometrial invasion <50% had no impact on patient outcomes [79].

The frequency rate of ovarian metastases ranges from 0.2–4% in cases treated with radical hysterectomy to 17–29% in autopsy series criteria, the latter higher percentages meaning the possible frequency of ovarian involvement in advanced cases [80,81]. However, limited data on the prognostic implications of adnexal involvement are still available in the literature.

Recently, L. Casey and N. Singh described two distinct metastasizing pathways in endocervical cancer: one related to HPVA tumors, pattern A-B sec. Silva and low-stage and one related to HPVI and high-stage tumors [82]. In detail, the former is characterized by an indolent and slow growth mainly by intraepithelial endometrial and tubal spread and/or transtubal exfoliation [82]. Differently, HPVI and high-stage tumors show higher incidence of LVSI, multinodular ovarian spread and a more aggressive behavior [82]. Therefore, the HPVA/low-stage group could be treated conservatively (ovarian sparing surgery), while the HPVI/high-stage group should be treated according to the FIGO stage and the other associated risk factors.

The prognostic impact of vaginal involvement from cervical cancer is well-known and is taken into account in the FIGO classification [1,4,74]. Cervical cancer may infiltrate vaginal tissue through contiguity, vessels permeation or both. The higher rates of vaginal invasion are observed in tumors measuring >20 mm and in adenocarcinomas, which frequently spread through vascular permeation [83]. According to FIGO 2018 classification, when cervical cancer is limited to the upper two-thirds of vagina without parametrial involvement, it is staged as Stage IIA [4,11]. When the disease involves the lower third of the vagina without reaching the pelvic walls, it is staged as Stage IIIA, which accounts for approximately 5% of patients [1,4,74]. Different stages mean different treatments. In fact, vaginal involvement by cervical cancer remains crucial for the therapeutic approach. In detail: stage IIA1 (neoplasm < cm 4) is usually treated with type C radical hysterectomy (with removal of upper vagina also); stage IIA2 (neoplasm > cm 4) needs type C radical hysterectomy plus adjuvant chemotherapy; stage IIIA requires concurrent chemo-radiation therapy [84,85].

Vaginal involvement may also be related to the cervical cancer pathogenesis. According to some studies, the vaginal microbiota plays an important role in the natural history of cervical cancer [84]. Mitra et al. described a possible link between CIN severity and vaginal microbiota: increasing disease severity is associated with the decreasing relative presence of Lactobacillus, suggesting the potential role in regulating Papillomavirus persistence and action [86].

Finally, four different patterns of recurrence in cervical cancer have been described: (i) endovaginal recurrence; (ii) paravaginal recurrence; (iii) invasion of surrounding organs (bladder, rectum and pelvic wall); (iv) vaginal recurrence associated with distant metastasis [87]. In a recent work, endovaginal recurrence and being RT naïve were independent factors for improved OS and PFS, in multivariate analyses [87].

### 4.2. Lymph Nodes Involvement

The size of pelvic lymph node metastasis is crucial in order to determine the N category according to the TNM staging system. In detail, metastases measuring greater than 2 mm are regarded as macrometastases and staged as pN1 [4]. Micrometastases measure greater than 0.2 mm but less than 2 mm and are staged as pN1(mi). Isolated tumor cells (ITCs) are single neoplastic cells or small clusters measuring not more than 0.2 mm in greatest dimension. ITCs do not upstage a carcinoma and are staged as pN0 (i+) [4]. Micrometastases show a negative impact on both the DFS and OS, whereas the real impact of ITCs remains unclear, and it is not known if ITCs can really degenerate in true lymph nodal metastasis [88].

The number of node metastases has been shown as a powerful prognostic predictor in cervical squamous cell carcinoma patients following radical surgery [89]. However, the cut-off values of the number of node metastases are slightly different across studies, ranging from 3 to 5 positive lymph nodes [89,90,91,92,93,94]. Regarding the total number of removed lymph nodes (RLNs), its prognostic and predictive value has been suggested by some recent studies [89,90,91,92,93,94]. However, other studies have failed to demonstrate its real prognostic significance; therefore, the role on RLNs remains controversial [89,90,91,92,93,94]. Moreover, excessive resection of negative nodes may increase the risk of postoperative complications. Imaging remains a useful diagnostic tool if pathological examination of lymph nodes cannot be achieved. However, the optimal size related to node positivity on MRI or CT is still debated [93]. The universally accepted criterion is that a short diameter of the lymph node less than 10 mm carries a good prognosis, while the prognosis significantly decreases in cases of short node diameter larger than 20 mm [93]. The anatomic location of nodal metastases is crucial for staging purposes since patients with pelvic node metastases are classified as stage IIIC1 and patients with para-aortic node metastases are classified as stage IIIC2; morever, para-aortic node metastases are recognized as an adverse prognostic factor [89,90,91,92,93,94].

For patients whose nodal metastases are limited to the pelvic cavity, several studies have demonstrated that multiple metastatic lymph nodes in the pelvic cavity carry a worse prognosis in comparison to a single metastatic node [91]. In addition, patients with common iliac nodal metastases carry a significantly reduced OS [91].

Metastatic Lymph Node Ratio (LNR) represents the percentage of metastatic nodes to total nodes retrieved and has recently emerged as a novel prognostic factor in cervical cancer [95]. According to recent studies, LNR represents a more accurate method to predict the OS of cervical cancer patients since the number and location of lymph nodes vary in each individual [95,96,97,98]. However, LNR has been demonstrated to not always reflect the real nodal tumor burden [95,96,97,98]. In this regard, in case of a limited nodal count, LNR may appear relatively high while a low LNR may be observed in more extensive nodal dissections. Therefore, LNR appears to be a reliable prognostic predictor only in cases where an adequate standard of lymphadenectomy is achieved, however, the minimum number of nodes that should be retrieved remains controversial [95,96,97,98].

Log Odds of Positive Lymph Nodes (LODDs) is a novel parameter that improves the accuracy of lymph node evaluation irrespective of nodal positivity status and has been identified as a powerful prognostic predictor in several malignancies, including cervical cancer, where a nomogram based on LODDS has showed good accuracy in predicting OS [99,100]. LODDs is defined as follows: log [(No. of positive lymph nodes + 0.5)/(No. of harvested lymph nodes–No. of positive lymph nodes + 0.5)] [99,100]. LODDs has the potential to distinguish different prognoses among patients with the same N stage. In this regard, if the number of nodal metastases is zero, the LNR will be zero; however, LODDs will vary according to the number of negative lymph nodes [99,100].

### 4.3. Distant Metastases

Metastatic cervical cancer carries an extremely poor prognosis and is often difficult to treat [101]. The reported median survival time of metastatic cervical cancer is 8–13 months, and the 5-year survival rate is 16.5% [101]. Due to the rarity of metastatic cervical cancer, a large-population based study is still lacking; therefore, the treatment strategies and the prognostic impact of the different metastatic sites are still poorly understood [101,102]. A recent paper by Yin et al. investigated the prognostic impact of metastatic sites in a cohort of 99 cervical cancer patients. His research revealed that liver metastases carried the worst prognosis [103]. These findings are different from previously published studies, which report similar survival times between different metastatic sites with a median survival of 9 months in lung metastasis, 7 months in liver metastasis, 6 months in brain metastasis and 8 months in bone metastasis [101,102,103,104]. Moreover, in the paper by Kim et al., patients with lung recurrences after treatment have shown a better prognosis in comparison to patients showing liver recurrences after treatment [104]. Another large-population-based study, including 1347 metastatic patients from the SEER database, demonstrated that the lung represented the most common metastatic site of cervical cancer; moreover, patients with a single metastatic site showed longer survival times in comparison to multi-site metastatic patients [102].

## 5. Molecular Markers and Future Perspectives

All relevant molecular prognostic markers are summarized in Table 4.

Several genetic alterations are reported in cervical cancer, and some of them are related to HPV-infection [105,106,107,108,109,110,111]. The progression from stage I to stage IV has been related to the loss of heterozygosity (LOH). Indeed, the loss of tumor suppressor protein caused by LOH is reported in both squamous cell carcinoma and adenocarcinoma of the cervix. The most frequent LOH loci are 3p14-22, 4p16, 5p1, 6p21-22, 11q23, 17p13.3, 18q12-2 and 19q13 [99,100]. Recently, The Cancer Genome Atlas (TCGA) database has expanded the knowledge of the genomic landscape of cervical cancer [107]. Based on molecular and integrative profiling, the following subgroups defined by different HPVs and molecular features were identified: keratin-low and keratin-high squamous cell carcinomas, adenocarcinoma-rich and endometrial-like cervical cancer. Most of squamous cell carcinomas belong to the keratin-low and keratin-high groups [107]. The endometrial-like subgroup emerged as HPV-negative tumors and showed a high frequency of KRAS, ARID1A and PTEN mutations. PIK3CA, PTEN and MPK1 were confirmed as significantly mutated genes in cervical cancer [101]. Several genes were found to be significantly mutated in cervical carcinoma, such as ERBB3, CASP8, HLA-A, SHKBP1 and TGFBR2. Notably, ERBB3 (HER3) may represent a novel therapeutic target. Among several amplifications, in the same study, they found a mutation in BCAR4 that is indirectly targetable by lapatinib and CD274 (PDL1) and PDCD1LG2 (PDL2) genes that may represent novel immunotherapy targets [101]. Interestingly, deregulated pathways, such as Wnt, PI3K/AKT/mTOR, VEGF, EGFR, Notch Hedgeog, significantly contribute to poor clinical outcome, reduced OS, distance metastasis chemo and radio recurrence [107]. All these pathways involve molecular inhibitors that could be useful for monitoring therapeutic response and target therapy. Moreover, recent evidence suggests a role of Lynch Syndrome, with MSI-high phenotype, also in cervical cancer [108]. Another recent study highlighted the role of MSI, PDL1 expression and Tumor Mutation burden in cervical cancer immunotherapy. In this study HPV-associated cervical cancer showed the highest frequency of PD-L1 expressions [109]. The role of PD-L and MSI as therapeutic biomarkers has also been demonstrated in neuroendocrine cervical cancer [110].

Finally, PD-L1 immunohistochemical expression with a CPS-score ≥1 is considered an eligibility criterium for pembrolizumab therapy in metastatic and recurrent squamous cell carcinoma [111].

## 6. Conclusions

In the present paper we aimed to review novel and potentially underestimated prognostic factors for which preliminary studies have shown promising results. In our opinion, LVSI, PNI, DOI, TFD, a novel grading system and TILs are the most relevant factors that may be included in the pathology report of cervical cancer in order to help future studies in validating and corroborating their suggested prognostic and predictive role.

## Figures and Tables

**Figure 1 cancers-15-01137-f001:**
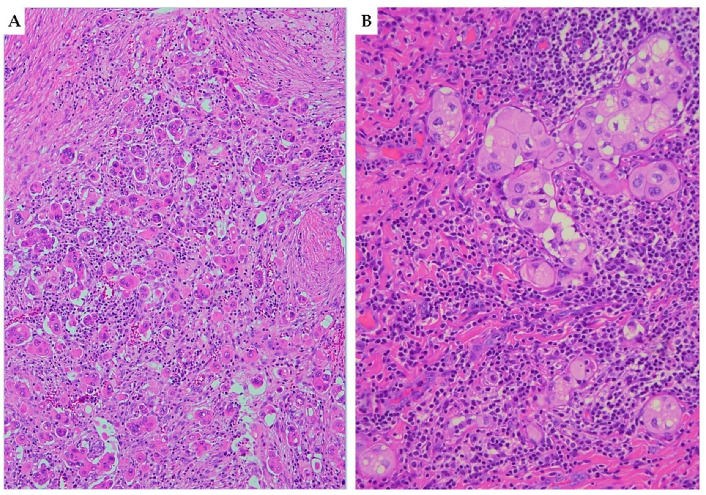
Haematoxylin and eosin (H&E) stained sections ((**A**) 10×; (**B**) 20×) illustrating tumor-budding and cell nest size in squamous cell carcinoma of the cervix. (**A**) Cervical squamous cell carcinoma with high tumor-budding activity, characterized by numerous small tumor clusters of <5 cells present at the infiltrating edge of the tumor (×10). (**B**) Small-sized cell nests: cervical cancer showing cell nests consisting of 2–4 tumor cells with nested architecture (×20).

**Figure 2 cancers-15-01137-f002:**
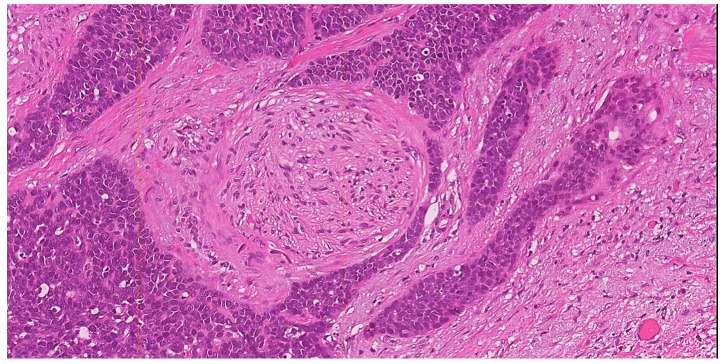
H&E stained section (10×) illustrating perineural invasion. In this example of squamous cell carcinoma of the uterine cervix with basaloid morphology, a small round nerve structure (center of the field) is surrounded by the neoplastic proliferation (×10).

**Table 1 cancers-15-01137-t001:** Prognostic factors related to primary tumors: squamous cell carcinoma.

Squamous Cell Carcinoma		
*Established* *Prognostic Factor*	*Novel Prognostic Factor*	*Uncertain Prognostic Utility*
HPV status	Tumor-budding/Cell nest size	Grading
Depth of stomal invasion LVSI Parametrial extension Margin status	Tumor-free distance (TFD) Perineural Invasion (PNI)	Horizontal extension
	TILS	

**Table 2 cancers-15-01137-t002:** Prognostic factors related to primary tumors: adenocarcinoma.

Adenocarcinoma		
*Established* *Prognostic Factor*	*Novel Prognostic Factor*	*Uncertain Prognostic Utility*
HPV status	Tumor-budding/Cell nest size	Grading Neuroendocrine differentiation Horizontal extension
Silva pattern of invasion		
Depth of stomal invasion LVSI	Tumor-free distance (TFD)	
Parametrial extension Margin status	Perineural Invasion (PNI)	
	TILS	
Special histologic types (gastric-type, clear cell, mesonephric, micropapillary, signet ring, invasive stratified mucinous carcinoma)		

**Table 3 cancers-15-01137-t003:** Prognostic factors related to advanced-stage tumors.

Squamous Cell Carcinoma/Adenocarcinoma		
*Established* *Prognostic Factor*	*Novel Prognostic Factor*	*Uncertain Prognostic Utility*
Size metastasis Number of metastatic lymph nodes	Metastatic Lymph Node Ratio	Endometrial extension
Location of metastatic lymph nodes	Log Odds of Positive LNs (LODDs)	Adnexal extension
Vaginal extension		
Distant metastases		

**Table 4 cancers-15-01137-t004:** Molecular prognostic markers.

Molecular Marker with Prognostic Utility
TCGA molecular subgroups keratin-low/keratin-high squamous cell carcinomas adenocarcinoma-rich endometrial-like
Therapeutic targets
Microsatellite instability
ERBB3 (HER3)
BCAR4
PDLI1
PDL2
Deregulated pathways Wnt PI3K/AKT/mTOR VEGF EGFR Notch Hedgeog

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
