# Peer review of "Recent Advances in Cervical Cancer Management: A Review on Novel Prognostic Factors in Primary and Recurrent Tumors"

_cancers, 2023, doi:10.3390/cancers15041137_

Round 1

Reviewer 1 Report

Santoro et al review the recent advances in the prognostic factors of primary-early and advanced, recurrent, or metastatic cervical carcinoma. The identification of new prognostic factors for cervical cancer is of great importance for researchers and practicing pathologists and oncologists. Despite the complexity of this field, the authors provide a rather comprehensive presentation of established and novel prognostic factors that may be included in pathology reports of early and advanced cervical cancer. However, the manuscript seems preliminary and has limitations that preclude publication at this stage. The main problem is that it is not clear which of the discussed prognostic factors have been already implemented in practice and which are novel and promising factors.

1. The introduction is incomplete. A question that needs to be answered in the introduction is about the current recommendations (by WHO or CAP) for pathology reports i.e which are the current essential and desired/recommended prognostic factors that need to be included in the pathology report?

2. Tables 1-2 need to be reorganized and to include additional information (histological subtype- squamous or adenocarcinoma, current status: established or novel prognostic factor, references)

3. What is the prognostic importance of HPV status, Silva pattern and horizontal extent of the carcinoma?

4. Lymph nodes in advanced cancer: Size criteria (micrometastasis and macrometastasis) are used for pN classification, what is their prognostic importance?

5. The manuscript needs significant language editing.

Author Response

We thank the reviewer for the time and effort dedicated to provide insightful feedback on our paper. Thus, it is with great pleasure that we resubmit our article for further consideration. We have incorporated changes that reflect the detailed suggestions you have graciously provided. To facilitate your review of our revisions, the following a point-by-point response to the questions and comments is provided. Please note that all changes to the original manuscript have been highlighted in yellow.

Q1. The introduction is incomplete. A question that needs to be answered in the introduction is about the current recommendations (by WHO or CAP) for pathology reports i.e which are the current essential and desired/recommended prognostic factors that need to be included in the pathology report?

A1. A novel section regarding the required/recommended items for pathology has been provided in the introduction.

Q2. Tables 1-2 need to be reorganized and to include additional information (histological subtype- squamous or adenocarcinoma, current status: established or novel prognostic factor, references)

A2: We introduced 3 new tables, synthetizing the current and the novel prognostic factors for both squamous cell cancer and adenocarcinoma.

Q3. What is the prognostic importance of HPV status, Silva pattern and horizontal extent of the carcinoma?

A3. We have written dedicated paragraphs in the text for these items.

Q4. Lymph nodes in advanced cancer: Size criteria (micrometastasis and macrometastasis) are used for pN classification, what is their prognostic importance?

A4. We added comments regarding size criteria for node metastases and its clinical significance

Q5. The manuscript needs significant language editing.

A5. We have checked the text and revised it.

Reviewer 2 Report

 In this short review on  novel prognostic factors in primary and recurrent tumors of the cervix, the authors suggest new parameters for  these. The manuscript has so many spelling errors and should be extensively proof read. 

Comments

I suggest HPV status should be  a major determinant of prognostic value and current molecular diagnostics are more than capable of distinguishing HPV positive Vs negative. 

Molecular prognostic markers could be elaborated in a table for more clarity.

The abstract, simple summary and introduction are very repetitive and uses the same sentences. This should be removed.

The brief section on CIN grading before primary tumor can be useful for the readers. Also, a figure showing tumor budding and cell nests and other parameters of cellular level may be more informative but not mandatory

Author Response

We thank the reviewer for the time and effort dedicated to provide insightful feedback on our paper. Thus, it is with great pleasure that we resubmit our article for further consideration. We have incorporated changes that reflect the detailed suggestions you have provided. Extensive language editing has also been performed. To facilitate your review of our revisions, the following a point-by-point response to the questions and comments is provided. Please note that all changes to the original manuscript have been highlighted in yellow.

Q1. I suggest HPV status should be  a major determinant of prognostic value and current molecular diagnostics are more than capable of distinguishing HPV positive Vs negative.

A1. We agree with the reviewer. We have written a dedicated paragraph in the text for this topic.

Q2. Molecular prognostic markers could be elaborated in a table for more clarity.

A2. We have elaborated a new table for molecular markers (Table 4)

Q3. The abstract, simple summary and introduction are very repetitive and uses the same sentences. This should be removed.

A3. We have corrected the two sections accordingly.

Q4. The brief section on CIN grading before primary tumor can be useful for the readers. Also, a figure showing tumor budding and cell nests and other parameters of cellular level may be more informative but not mandatory

A4. We have added a brief section for CIN grading and two figures illustrating tumor budding, cell nest size and perineural invasion.

Round 2

Reviewer 1 Report

The authors addressed the reviewers' points in a very satisfactory way. The current manuscript is much better and should be published after minor editing changes.

- Line 65 b?

- Please mention magnification in the figure legends

- Please correct a few typos, still present in the manuscript.

Author Response

Thanks for your positive comments. We provided our point-by-point response to your comments and advices

- Line 65 b?

- We have corrected this line

- Please mention magnification in the figure legends

- We have done

- Please correct a few typos, still present in the manuscript.

- The text has been revised

Reviewer 2 Report

The authors have responded adequately to my comments and included more information. I have no further comments.

Author Response

Thank you for your positive comments.